# Astrocyte-Mediated Plasticity: Multi-Scale Mechanisms Linking Synaptic Dynamics to Learning and Memory

**DOI:** 10.3390/cells14241936

**Published:** 2025-12-05

**Authors:** Masaya Yamamoto, Tetsuya Takano

**Affiliations:** 1Kyushu University Institute for Advanced Study, Kyushu University, 3-1-1 Maidashi, Higashi-ku, Fukuoka 812-8582, Japan; masaya.yamamoto@bioreg.kyushu-u.ac.jp; 2Division of Molecular Systems for Brain Function, Kyushu University Medical Institute of Bioregulation, 3-1-1 Maidashi, Higashi-ku, Fukuoka 812-8582, Japan; 3PRESTO (Precursory Research for Embryonic Science and Technology), Japan Science and Technology Agency, 4-1-8 Honcho, Kawaguchi, Saitama 332-0012, Japan

**Keywords:** astrocyte-mediated plasticity, tripartite synapse, astrocyte calcium dynamics, learning and memory

## Abstract

Astrocytes play a pivotal role in shaping synaptic function and in learning, memory, and emotion. Recent studies show that perisynaptic astrocytic processes form structured interactions with pre- and postsynaptic elements, which extends synaptic diversity beyond neuron–neuron connections. Accumulating evidence indicates that astrocytic Ca^2+^ signaling, gliotransmission, and local translation modulate synaptic efficacy and contribute to the formation and stabilization of memory traces. It is therefore essential to define how astrocytic microdomains, multisynaptic leaflet domains, and network-level ensembles cooperate to regulate circuit computation across space and time. Advances in super-resolution and volumetric in vivo imaging and spatial transcriptomics now enable detailed, cell-type- and compartment-specific analysis of astrocyte–synapse interactions in vivo. In this review, we highlight these approaches and synthesize classical and emerging mechanisms by which astrocytes read neuronal activity, write to synapses, and coordinate network states. We also discuss theoretical frameworks such as neuron–astrocyte associative memory models that formalize astrocytic calcium states as distributed substrates for storage and control. This integrated view provides new insight into the multicellular logic of memory and suggests paths toward understanding and treating neurological and psychiatric disorders.

## 1. Introduction

Astrocytes are now recognized as active partners in synaptic function rather than passive support cells. Perisynaptic astrocytic processes actively sense neurotransmitter release, generate intracellular Ca^2+^ signals [1], and regulate the synaptic milieu through gliotransmission, ion buffering, glutamate uptake, and metabolic coupling [2,3,4,5]. Within the tripartite organization that includes the presynaptic terminal, the postsynaptic spine, and the surrounding astrocyte, astrocytes modulate classical forms of plasticity such as long-term potentiation (LTP), long-term depression (LTD), and spike-timing-dependent plasticity (STDP) [6,7,8], which expands plasticity beyond a neuron-only phenomenon to a neuron–glia interaction [9,10,11]. Recent advances in super-resolution microscopy [12], genetic manipulation, and computational analysis indicate strong spatial and temporal compartmentalization. Nanoscale calcium microdomains (also called “hotspots” or “calcium nanodomains”) are very small areas where calcium levels rise inside the thin perisynaptic processes. These localized calcium signals allow astrocytes to affect each synapse in a specific way. Interconnected leaflet networks (also called peri-synaptic astrocytic processes, PAPs), which are the smallest terminal branches of astrocytes, seem to combine or integrate activity coming from many different synapses [13,14,15,16]. At larger scales, ensembles of co-activated astrocytes have been shown to participate directly in memory encoding and recall, indicating that astrocytes themselves may form functional components of the memory engram [17].

It is therefore timely to reassess the long-standing neuron-centric view of learning and memory. Since Hebb’s postulate, most models and experiments have framed plasticity as changes in synaptic weights within neuronal networks [18,19,20,21,22]. This framework has been foundational. However, classical neuron-only models show clear limitations. Memory capacity tends to scale only linearly with network size, intrinsic homeostatic safeguards remain underspecified, and millisecond electrical rules are difficult to reconcile with slower biochemical and metabolic processes that support durable behavioral change [23,24,25,26,27,28]. The tripartite synapse concept, together with accumulating evidence for astrocyte-dependent modulation, introduces additional state variables, time constants, and control mechanisms that can address these gaps [1,2,3,4,5,6,7,8,9,29,30].

Theoretical work has begun to formalize astrocytes as computational elements. Emerging frameworks, exemplified by the NAAM model, propose that information can be stored not only in neuronal synaptic weights but also in dynamic Ca^2+^ states within astrocytic processes [31,32,33,34]. These states may enhance capacity, temporal stability, and context-dependent modulation. Although these predictions require further experimental testing, they offer a principled bridge between biological computation and artificial architectures and motivate targeted experiments on astrocyte–neuron cooperation [35,36].

In this review, we integrate classical and emerging perspectives on astrocyte-mediated synaptic plasticity and computation across scales. We first revisit the foundational mechanisms within the tripartite synapse and clarify their limitations. We then move from nanoscale Ca^2+^ microdomains and leaflet networks to mesoscale astrocyte ensembles that are implicated in learning and recall. By synthesizing experimental and theoretical insights, we highlight how astrocytes contribute to memory storage, network coordination, and computational capacity, fundamentally redefining how the brain learns and remembers (Figure 1 and Table 1).

## 2. Literature Search Strategy and Inclusion Criteria

To guide a comprehensive and balanced narrative synthesis and to enhance the transparency of our literature scope, we conducted a structured search of studies published between January 1990 and October 2025. Searches were performed in PubMed, Scopus, and Web of Science using combinations of the terms “astrocyte”, “synaptic plasticity”, “calcium signaling”, “tripartite synapse”, “astrocyte ensemble”, and “learning and memory”. Additional studies were identified by examining the reference lists of key primary papers and recent reviews.

The focus of this narrative review is on mammalian central nervous system synapses, integrating mechanistic, physiological, and computational perspectives. We do not aim to provide exhaustive coverage of all glial synapses, nor do we systematically review peripheral tripartite synapses such as those at the neuromuscular junction, where glial modulation is also well established (reviewed in [37,38]).

In the neuron-centric model, plasticity arises from pre- and postsynaptic mechanisms such as NMDA receptor activation and AMPAR trafficking, with memory stored at individual neuronal synapses. In the classical tripartite model, plasticity involves coordinated activity among presynaptic terminals, postsynaptic neurons, and astrocytic leaflets, where astrocytic Ca^2+^ signaling and gliotransmission modulate synaptic strength and timing. In the emerging integrative model, astrocytes contribute to plasticity through multi-synaptic and cellular-scale calcium states that integrate network activity over time, making astrocyte microdomains an additional site for distributed memory encoding and storage within neural circuits.

## 3. Classical Core Mechanisms of Astrocyte-Mediated Synaptic Plasticity

Ben Barres’s pioneering studies established astrocytes as powerful drivers of synaptogenesis and synaptic efficacy [39], and on this foundation Araque et al. (1999) formalized the tripartite synapse [29], reframing plasticity to include astrocytes as active regulators rather than passive support cells. In this framework, perisynaptic astrocytes closely ensheath most excitatory synapses in the hippocampus [40] and cortex [41,42], enabling them to sense [30], integrate [30,43], and modulate [16,44] neuronal activity with remarkable precision [13]. Classical models propose that astrocytes may contribute to synaptic plasticity through several mechanisms, including (1) calcium-dependent release of neuroactive molecules, (2) extracellular homeostasis, and (3) metabolic support.

A central proposed mechanism is gliotransmission, the Ca^2+^-dependent release of neuroactive molecules from astrocytes [45,46]. Activation of metabotropic receptors such as mGluRs [47], GABA_B [48], and P2Y [49] triggers intracellular Ca^2+^ elevations, leading to the release of glutamate [50], ATP [51], D-serine [52], or GABA [53]. These gliotransmitters fine-tune synaptic efficacy: D-serine acts as an NMDA receptor co-agonist required for LTP induction [52,54], ATP-derived adenosine suppresses presynaptic release [55,56]. Astrocyte-derived glutamate or GABA [57] can, respectively, enhance or dampen neuronal excitability. Through these processes, astrocytes exert bidirectional control over synaptic strength and plasticity [58,59]. While these findings support a role for gliotransmission, the physiological relevance of gliotransmission in vivo remains actively debated [60,61]. Studies using IP_3_R2 knockout mice, where global astrocytic Ca^2+^ elevations are disrupted, show largely preserved LTP and complex behavior [62,63]. These observations raise questions about how frequently gliotransmission participates under physiological conditions, although they may also reflect compensatory Ca^2+^ signaling through alternative pathways such as IP_3_R1/3, store-operated calcium entry, or microdomain-restricted Ca^2+^ events that remain intact in IP_3_R2-deficient astrocytes [64]. Concerns have also been raised regarding the cellular origin of D-serine [65,66], and vesicular versus channel-mediated release pathways [67], indicating that gliotransmission is likely a context-dependent mechanism [64].

Astrocytes also maintain the extracellular milieu necessary for stable transmission. High-affinity glutamate transporters GLT-1 and GLAST rapidly clear glutamate from the cleft, limiting receptor overactivation and preventing excitotoxicity [68]. Inwardly rectifying potassium channels (Kir4.1) and extensive gap-junction coupling enable redistribution of K⁺ ions, preventing local hyperexcitability [69]. Moreover, astrocytes sustain neuronal metabolism via the astrocyte–neuron lactate shuttle, ensuring that energy supply matches the demands of activity-dependent plasticity [70,71]. Furthermore, astrocytes secrete matricellular proteins such as thrombospondins (TSP1/2), hevin (SPARCL1), and SPARC. Thrombospondins promote excitatory synapse formation [72], while hevin facilitates excitatory synaptogenesis and SPARC antagonizes this process by competing for shared binding partners [73,74]. In addition, central neuropeptides such as oxytocin, vasopressin, and CRH act as potent modulators of astrocytic calcium signaling, metabolism, and gene expression, linking astrocyte function to mood regulation and neurodegenerative disease mechanisms [75].

The induction and maintenance of classical forms of synaptic plasticity such as LTP, LTD, and STDP. The timing of gliotransmitter release can alter the polarity of spike-timing-dependent plasticity [6,8], while astrocytic p38α MAPK signaling has been shown to be essential for hippocampal LTD expression [76,77]. By regulating neurotransmitter availability, receptor occupancy, and ionic balance, astrocytes actively shape the rules of synaptic modification [30,45,78].

Nevertheless, the temporal mismatch between astrocytic Ca^2+^ elevations (seconds to minutes) and fast synaptic events (milliseconds) raises questions about how astrocytes interact with rapid forms of synaptic plasticity [79,80]. Similarly, synaptic specificity remains a major unresolved issue: broad astrocytic Ca^2+^ transients could impact many synapses simultaneously unless constrained by microdomain-level compartmentalization, which was not resolved by early imaging studies [81]. Moreover, the model treated astrocytes as homogeneous units, overlooking their regional diversity [82] and nanoscale structural specialization [83]. While classical mechanisms explain modulation of local synapses, they do not fully account for how astrocytes coordinate information across networks or sustain long-term plasticity over extended timescales [84,85].

These limitations have collectively prompted a conceptual expansion beyond the classical framework. Advances in high-resolution imaging, genetic manipulation, and modeling now reveal a hierarchically organized architecture of astrocytic signaling. Structures from nanoscale microdomains to network-level ensembles support integrative and dynamic roles in learning and memory, as discussed in the following sections.

## 4. Beyond the Classical Tripartite Model

Recent discoveries have expanded our understanding of astrocytes far beyond the boundaries of the classical tripartite synapse [78,86]. Rather than functioning as uniform modulators, astrocytes exhibit highly compartmentalized organization and operate across multiple spatial and temporal scales. At the nanoscale, fine perisynaptic processes form distinct Ca^2+^ microdomains that decode synapse-specific signaling [13,15,87]. At larger scales, astrocytic leaflets and process networks integrate activity across thousands of synapses, enabling context-dependent modulation of neuronal ensembles [14,83].

Astrocytes also display remarkable computational versatility. They can decode neuronal firing patterns [2,88], release gliotransmitters with temporal precision [67], and coordinate large-scale network oscillations [89,90]. Emerging evidence further indicates that subsets of astrocytes participate directly in memory encoding and recall, forming astrocytic ensembles in memory that influence behavioral output [17,91].

In the following sections, we examine how this hierarchical organization from microdomains to network-level assemblies [92] endows astrocytes with integrative and computational capabilities that extend well beyond the classical neuron-centric view of plasticity [93,94] (Figure 2 and Table 2).

Schematic illustration of the hierarchical organization of astrocyte signaling and integration across spatial and functional scales. At the single tripartite synapse level, individual tripartite synapses are modulated by localized Ca^2+^ transients and astrocytic read–write mechanisms. At the multi-synaptic level, a single astrocytic leaflet interacts with up to ten synapses, integrating local inputs into coherent microcircuit modulation. At the multi-leaflet level, neighboring leaflets form gap–junction-interconnected domains that coordinate calcium activity across a single astrocyte territory, thereby supporting domain-wide information processing. At the single-astrocyte level, one protoplasmic astrocyte contacts up to 10^6^ synapses, integrating excitatory and inhibitory inputs through intracellular calcium signaling and metabolic coupling. At the network level, intercellular communication via Ca^2+^ waves through gap junction coupling, and gliotransmitter release synchronizes astrocytic and neuronal ensembles across brain regions. Finally, at the ensemble level, distributed astrocyte populations form functional networks that participate in learning, memory processing, and state-dependent coordination of neural circuits. Adapted from [14].

### 4.1. Microdomains and Leaflet Domains: Hierarchical Units of Astrocytic Integration

Recent studies revealed that astrocytic signaling is executed through a hierarchically organized nanoscale architecture that supports both local precision and mesoscale integration [80,83]. Within this framework, two complementary structural and functional units emerge: microdomains, which mediate Ca^2+^ signaling at single synapses [13], and leaflet domains, which integrate activity across multiple neighboring synapses [14]. Together, these levels of organization endow astrocytes with the capacity to decode, transform, and coordinate information within complex neuronal networks.

#### 4.1.1. Microdomains: Synapse-Specific Read–Write Units

Super-resolution 3D-STED microscopy has shown that fine astrocytic processes form a spongiform meshwork composed of bulbous nodes connected by narrow shafts which are thin, elongated tubular structures. Each node, typically 300–400 nm in diameter, contacts dendritic spines and contains endoplasmic reticulum fragments enriched in IP_3_ receptors [13]. This node–shaft alternation imposes diffusion barriers, generating discrete biochemical compartments where Ca^2+^ transients remain spatially confined [87]. Within these microdomains, Ca^2+^ transients are triggered by synaptic glutamate spillover [118,119], ATP release, or neuromodulator action via metabotropic receptors [1]. These localized events are fast (hundreds of milliseconds to seconds) and asynchronous across the astrocyte, enabling the simultaneous monitoring of thousands of synapses with high spatial fidelity [15].

Astrocytes do not merely relay this synaptic information [111]. Covelo and Araque [2] demonstrated that single astrocytes can distinguish neuronal firing patterns and release distinct combinations of gliotransmitters. Short or low-frequency stimulation (~1 s or ≤20 Hz) elicits Ca^2+^-dependent glutamate release and transient potentiation, whereas high-frequency activity (30–90 s or >20 Hz) produces biphasic effects, with an initial potentiation followed by ATP/adenosine-mediated depression. Such pattern-specific responsiveness enables bidirectional regulation of synaptic strength [120], filtering transient versus sustained neuronal activity [45]. Microdomains thus perform synapse-specific read and write operations, converting electrical activity into biochemical modulation at the single-synapse level [96,121].

At the molecular level, astrocytic local translation represents an important molecular mechanism enabling calcium-dependent modulation of tripartite synapses. Recent studies using PAP-TRAP (peripheral astrocyte processes-translating ribosome affinity purification) and puromycin labeling [98] have revealed that astrocytes actively synthesize proteins within their perisynaptic processes, enabling rapid and spatially precise regulation of local proteomes. Ribosomes and mRNAs are abundant in these fine processes, and Ca^2+^ signals triggered by synaptic activity locally activate translation via mTOR [122] and eIF2α pathways [101]. This activity-dependent protein synthesis produces neurotransmitter transporters (GLT-1, GLAST), metabolic enzymes (Glul), and synapse-regulating factors (Thbs4, Hevin, SPARC, MERTK), allowing astrocytes to remodel individual microdomains in real time in response to neuronal activity. Recent evidence further demonstrates that astrocytic translation is essential for memory consolidation, as contextual fear conditioning modulates both the distribution and translation of ribosome-bound mRNAs within perisynaptic astrocytic processes [99]. Several PAP-enriched transcripts—such as *Flt1*, *Fth1*, *Ccnd2*, *Mdm2*, *Gnb2l1*, and *Eef1a1*—are directly linked to learning and memory [123,124,125], suggesting that local translation dynamically reconfigures perisynaptic proteomes to support long-term synaptic plasticity and memory stabilization.

#### 4.1.2. Leaflet Domains: Structural Basis for Multi-Synaptic Integration

Beyond the single-synapse scale, astrocytic leaflet domains represent a newly recognized level of organization [107] that enables multi-synaptic integration. Recent volumetric electron microscopy and nanoscale calcium imaging [14] revealed that leaflets are ultra-thin lamellar extensions (<250 nm) emanating from distal processes. Quantitative reconstructions show that astrocytic leaflets ensheath approximately 90% of excitatory synapses in clusters and only ~10% individually, indicating that astrocytes regulate groups of synapses collectively rather than in isolation. Each leaflet contains minute endoplasmic reticulum (ER) saccules expressing IP_3_R1 [97,126], but notably lacks mitochondria, suggesting specialization for rapid, transient calcium signaling. Crucially, leaflets are interconnected via gap junction connexin43 [14], creating cytosolically continuous domains through which calcium and other signaling molecules propagate [104,127]. This organization allows multiple leaflets to function as integrated computational units capable of coordinating responses across clusters of synapses. In these domains, signals from several presynaptic neurons converge, permitting astrocytes to act as spatiotemporal integrators of network activity rather than passive modulators of single synapses.

Calcium signaling within leaflet domains follows threshold-dependent dynamics [92].Physiological increases in intracellular Ca^2+^ (~80–140 nM) are sufficient to trigger localized glutamate release [112], whereas larger or sustained elevations recruit additional gliotransmitter pathways such as ATP and D-serine release [128]. These graded responses convert variations in input strength and duration into specific biochemical outputs, enabling astrocytes to filter or amplify neuronal activity in a context-dependent manner. When multiple neighboring synapses are co-activated, calcium transients originating in individual leaflets can merge into prolonged, high-amplitude waves, reflecting an emergent property of cooperative integration within the leaflet network.

Functionally, leaflet domains serve as biochemical control hubs that bridge microdomain precision with mesoscale coordination. By pooling inputs from roughly ten or more presynaptic neurons, they stabilize excitation–inhibition balance [129] while preserving synapse-specific modulation. The architecture of leaflet domains thus provides a biophysical substrate for analog summation and threshold gating, analogous to nonlinear operations in computational systems [130]. This mechanism may underlie higher-order processes such as memory formation, decision-making, and contextual learning [85,116], in which selective activation and suppression of synaptic ensembles are required for adaptive behavior. Collectively, the integration of microdomain-level decoding and leaflet-level computation supports a unified view of astrocytes as hierarchical processors. Through localized calcium events and domain-level coupling, astrocytes transform distributed neuronal activity into coherent network-level modulation, linking microscopic signaling with macroscopic cognition. To provide quantitative context for these hierarchical structures, we summarize characteristic dimensions, calcium signal kinetics, and synaptic coverage ratios across microdomains, leaflets, single-cell territories, and network scales in Table 3. This table integrates values reported across super-resolution imaging, in vivo calcium imaging, and anatomical reconstruction studies.

### 4.2. Astrocytes as Network Coordinators

Beyond their nanoscale precision, astrocytes operate as large-scale integrators that coordinate neuronal activity across thousands of synapses [133]. Each protoplasmic astrocyte in the cortex or hippocampus occupies a distinct territorial domain encompassing tens to hundreds of thousands of synapses [108]. This organization enables astrocytes to detect distributed patterns of neuronal activity and transform them into coherent modulatory outputs, positioning them as essential computational hubs within neural circuits [134]. Astrocytes also exhibit regional and structural diversity—cortical astrocytes emphasize glutamatergic modulation [83,102], whereas white-matter astrocytes specialize in metabolic and axonal support [110,135]. Moreover, recent studies reveal that astrocytes regulate circuits not only through classical secreted factors but also via contact-mediated adhesion mechanisms [136,137], providing spatial precision and dynamic control of synaptic organization.

#### 4.2.1. Network-Level Coordination

Astrocytic calcium signaling extends beyond local synaptic modulation to orchestrate large-scale network states. In vivo two-photon imaging demonstrates that astrocytic Ca^2+^ transients can synchronize previously asynchronous neuronal populations, generating coherent oscillations essential for attention, sensory binding, and memory consolidation [109,138]. Manipulation of astrocytic calcium activity alone is sufficient to induce transitions between synchronous and asynchronous cortical states [90,109], highlighting their causal role in governing network-level excitability and rhythmic organization.

Astrocytes also play a crucial role in maintaining the excitatory–inhibitory balance necessary for stable computation. Through temporally and spatially precise release of glutamate, ATP, and D-serine, they stabilize recurrent network activity and prevent runaway excitation [45,129]. Both experimental and computational studies indicate that astrocyte-modulated tripartite synapses can regulate oscillatory coherence [89], especially within recurrent feedback loops that sustain rhythmic network activity [30,139]. In this way, astrocytes function as adaptive homeostatic controllers [140,141] that balancing excitability and inhibition to maintain circuit stability while permitting experience-dependent plasticity.

#### 4.2.2. Temporal Integration and Metaplasticity

Astrocytes supply a slow and integrative timescale that enriches neural computation. Their calcium waves can summate inputs over seconds to minutes, bridging transient synaptic activity with long-term modifications in circuit function [142]. This capacity allows astrocytes to convert rapid neuronal signals into persistent modulatory states that contribute to metaplasticity, the plasticity of plasticity itself [32,106]. Astrocyte-mediated gliotransmission regulates oscillatory coupling and temporal precision across cortical and subcortical circuits. Astrocytic glutamate release enhances gamma oscillation power (30–50 Hz) during active cognitive states [89], while ATP- and adenosine-dependent modulation of inhibitory interneurons improves oscillatory stability and synchrony [143,144,145]. These findings reveal that astrocytes act as temporal integrators, synchronizing neuronal ensembles across multiple timescales and linking fast synaptic signaling to slower, modulatory rhythms that underlie cognition and behavioral adaptation.

#### 4.2.3. Region-Specific Coordination and Adaptive Plasticity

Astrocyte-mediated network coordination is further shaped by regional diversity in molecular signatures and circuit connectivity. Transcriptomic analyses show that astrocytes express distinct gene programs depending on their brain region, developmental origin, and local neuronal partners [146,147,148]. This regional specialization enables astrocytes to differentially regulate excitatory and inhibitory transmission [135,149], neuromodulator sensitivity [150], and oscillatory dynamics across cortical, hippocampal, and subcortical networks [89,109,146].Functionally, this diversity allows astrocytes to implement region-specific computational strategies. For example, astrocytes in the hippocampus contribute to theta–gamma coupling crucial for memory encoding [113], while those in the prefrontal cortex modulate sustained firing patterns associated with working memory and decision-making [151]. White matter astrocytes, in contrast, emphasize metabolic homeostasis and long-range axonal conduction rather than rapid synaptic modulation [110]. Under pathological conditions, this heterogeneity becomes more pronounced: Serrano-Pozo et al. [152] demonstrated region- and stage-specific transcriptomic remodeling of astrocytes in Alzheimer’s disease, while Clayton et al. [153] identified injury- and disease-specific reactive subtypes that influence circuit recovery and degeneration. Recognizing such diversity reframes astrocyte-mediated network control not as a uniform process but as a regionally tuned and functionally adaptive system. Through this multiscale organization, from local microdomain signaling to domain wide integration and global network synchronization, astrocytes enable the brain to maintain stability, flexibility, and computational depth across diverse behavioral contexts.

#### 4.2.4. Adhesion-Based Mechanisms for Astrocytic Control of Neural Circuits

Astrocytes not only modulate neural activity through diffusible factors such as thrombospondins [72] and hevin [73] but also engage in direct contact-dependent adhesion mechanisms that shape synaptic connectivity and circuit function. These adhesion-based interactions represent an emerging paradigm [154] in which astrocytes use cell adhesion molecules (CAMs) to establish precise physical interfaces with neurons [136], enabling spatially restricted, bidirectional signaling [74] essential for synapse formation, maintenance, and plasticity [155].

Astrocytic neuroligins (NL1–3) and neurexins form trans-synaptic complexes that synchronize astrocyte morphological maturation with excitatory and inhibitory synaptogenesis during postnatal development [136]. NrCAM, acting through homophilic adhesion with neuronal NrCAM and coupling to the postsynaptic scaffold gephyrin, organizes GABAergic synapses and constrains astrocyte process overgrowth into the neuropil [156,157]. Integrin-based focal adhesion complexes (β-integrin, Talin, FAK, Tensin) further regulate astrocyte coverage of synapses and glutamate transporter expression, maintaining synaptic homeostasis under dynamic network conditions [158]. N-cadherin/δ-catenin complexes coordinate layer-specific astrocyte morphogenesis and neuronal dendrite stabilization [159] while Eph-ephrin signaling governs synapse remodeling, elimination, and astrocyte reactivity in development and disease [160,161,162]. Moreover, astrocytic phagocytic receptors MEGF10 and MERTK mediate selective synapse pruning, providing circuit-specific refinement of excitatory and inhibitory connectivity [163,164]. Collectively, these adhesion pathways operate in two modes. In a cooperative mode, where astrocyte–neuron adhesions promote synaptic maturation and stability, and in a competitive mode, where molecules like NrCAM function as “stop signals” to prevent excessive astrocytic infiltration into synaptic territories [156,157]. Dysregulation of these mechanisms can lead to excitatory–inhibitory imbalance and contribute to neurodevelopmental and psychiatric disorders such as autism and schizophrenia [136,165,166]. Thus, adhesion-based signaling establishes astrocytes as structural and molecular architects of neural circuits—integrating physical adhesion with biochemical communication to coordinate circuit assembly, balance, and long-term plasticity.

### 4.3. Astrocyte Ensembles in Memory Regulate Memory Processing

An emerging paradigm reshaping the neuroscience of memory is that astrocytes form functional ensembles that regulate the storage and recall of memories. These astrocyte ensembles in memory consist of spatially clustered astrocytes co-activated during memory processing [17,91]. Their coordinated activation suggests that astrocytes may contribute directly to the formation, maintenance, and retrieval of memory traces—beyond a purely supportive or modulatory role.

#### 4.3.1. Early Evidence for Astrocytic Contributions to Memory

Initial studies demonstrated that activating astrocytes during learning enhances memory formation. Chemogenetic stimulation of hippocampal astrocytes during fear conditioning using Gq DREADDs strengthened synaptic potentiation and improved long-term memory [116]. Astrocyte-neuron lactate transport was shown to be required for long-term memory consolidation [167]. Subsequent calcium imaging studies revealed task-specific astrocytic activation patterns during learning and recall across multiple regions, including the hippocampus [116], amygdala [168], and prefrontal cortex [15,16]. Together, these findings established that astrocytes dynamically participate in memory-related network activity.

#### 4.3.2. Astrocytic Ensembles as Memory Engram Components

A landmark study by Williamson, Kwon, and colleagues [17] provided direct evidence that astrocytes form engram-like ensembles essential for memory storage and recall. Learning experiences induced c-Fos expression in approximately 2–4% of hippocampal astrocytes, which were selectively reactivated during recall. These memory-related astrocytic ensembles, termed learning-associated astrocyte ensembles (LAAs), were spatially clustered near engram neurons, exhibited coordinated calcium dynamics, and enhanced local synaptic connectivity. Functional manipulations confirmed their causal role. Chemogenetic activation of astrocytic ensembles in memory, using a c-fos promoter–CreER–Lox-Stop-Lox intersectional genetic system, reactivated associated neuronal engrams and induced context-independent memory recall, whereas astrocyte-specific Fos deletion disrupted learning and recall. Transcriptomic analyses identified NFIA as a key transcriptional regulator of ensemble formation, with its loss impairing astrocyte-driven modulation of neuronal excitability. Recently, AstroLight technology, a calcium-dependent and light-activated gene expression system labeled with fluorescent markers, enables selective identification and manipulation of behaviorally relevant astrocytes in the nucleus accumbens [115]. Reactivation of these astrocytic ensembles drove reward-seeking behavior even in the absence of external cues. According to Dewa et al. [91], astrocytic ensembles stabilize long-term memory through sequential priming and activation. Learning induces adrenergic receptor upregulation that primes the ensembles. Subsequent recall drives cAMP and Ca^2+^ signaling and IGFBP2 release, which strengthens local engrams. These observations indicate that astrocytic ensembles may serve as multi-day eligibility traces for long-term stabilization. Collectively, these discoveries redefine the concept of the memory engram as a multicellular phenomenon. Astrocytic ensembles encode memory-relevant information in parallel with neurons, coordinate local plasticity, and regulate recall through calcium-dependent and transcriptional mechanisms. This neuron–astrocyte partnership represents a fundamental shift toward an integrated model of memory formation and stabilization in the mammalian brain.

### 4.4. Astrocytic Computational Frameworks and the Neuron–Astrocyte Associative Memory (NAAM) Model

Recent experimental discoveries have decisively shifted the perception of astrocytes from passive support cells to active computational partners in memory processing [78,117,169]. In parallel, theoretical neuroscience has begun to formalize this idea, positioning astrocytes as core elements within hybrid neuron–glia computational frameworks. Early conceptual models proposed that astrocytes could transiently store or integrate information through ionic or molecular states. Caudle [170] suggested that astrocytic ion-channel configurations might serve as a substrate for “glial memory,” while the “Memory Orchestra” framework [117,171] posited that astrocytes encode contextual aspects of experience such as spatial and temporal context through lasting biochemical states. Experimental support for these ideas came from Williamson et al. [17], who demonstrated that learning-associated astrocyte ensembles are reactivated during memory recall, providing direct evidence that astrocytic populations may participate in mnemonic encoding. Together, these findings inspired a new generation of computational models that extend beyond neuron-centric paradigms.

Building on these discoveries, the Neuron–Astrocyte Associative Memory (NAAM) model proposed by Kozachkov, Slotine, and Krotov [93] provides the first rigorous theoretical framework describing astrocytic computation. NAAM posits that memories are encoded not only in neuronal synaptic weights but also in the dynamic calcium signaling states between astrocytic processes. The state of each process is defined by its bidirectional interactions with neurons at tripartite synapses and its lateral communication with neighboring processes through intracellular calcium transport. Each process therefore acts as a semi-independent computational subunit, while coordinated calcium dynamics across multiple processes give rise to stable, distributed attractor states that could maintain information beyond transient neuronal firing.

Recent evidence from Lucas Benoit et al. [14] supports a plausible biological substrate for this interprocess communication. Their study revealed that astrocytic leaflets are interconnected by connexin43-mediated gap junctions, forming cytosolically continuous domains that permit calcium and IP_3_ to propagate between adjacent processes [172]. This organization allows localized calcium events to merge into broad, long-lasting elevations, consistent with the collective attractor dynamics envisioned in the NAAM model. However, connexin43 coupling alone is unlikely to fully account for intracellular calcium transport. Additional mechanisms may also support lateral propagation of signals between astrocytic processes. ER-based calcium diffusion [95], mitochondrial buffering [173,174], and local store-operated calcium entry (SOCE) [175] collectively enable coherent, memory-like calcium patterns [92,105]. Within this framework, astrocytes function not as simple modulators but as distributed computational arrays. Each process encodes a local calcium state that reflects recent neuronal activity [13], while process-to-process interactions integrate these local representations into a coherent astrocytic state [14]. The resulting calcium dynamics can persist for seconds to minutes [16,142], far longer than millisecond neuronal firing, and thus could provide a plausible mechanism for activity-silent memory, the temporary maintenance of information without sustained electrical activity. This property suggests that astrocytes may act as intermediate memory buffers that preserve contextual information across time gaps in neuronal signaling [34,176]. Mathematically, the NAAM model predicts supralinear memory capacity (K_max_ ∝ N^3^), dramatically exceeding the linear scaling limit of classical Hopfield networks [177,178]. This enhancement arises from the astrocytic layer’s ability to couple large numbers of synapses through slow calcium-based feedback, effectively expanding the dimensionality of the representational space available for memory storage. A complementary theoretical study by Kozachkov et al. [179] further demonstrated that tripartite synapses can perform normalization operations analogous to the “self-attention” mechanism in Transformer architectures [180,181], linking astrocyte–neuron interactions to attention-like computation observed in artificial intelligence systems. These ideas remain theoretical and require substantial biological validation.

Functionally, this hybrid neuron–astrocyte computation may endow the brain with several key advantages. First, astrocytes provide temporal stability, bridging fast neural activity with slower biochemical integration. Second, they introduce nonlinear, context-dependent modulation, allowing networks to dynamically adjust learning rules according to ongoing physiological state. Third, they enable distributed redundancy, as astrocytic calcium states can maintain information even when neuronal activity ceases, ensuring robustness against transient disruptions. Together, these properties suggest that astrocytes play a fundamental role in maintaining the brain’s high-capacity, stable, and adaptive memory systems. However, whether astrocytic calcium states operate with sufficient specificity, speed, and reliability to support these computations in vivo remains an open question.

The NAAM model also generates clear experimental predictions. Disrupting calcium diffusion within astrocytes through pharmacological blockade of gap junctions [182] or genetic deletion of connexin43 [103] would be degrade memory consolidation and recall by fragmenting astrocytic attractor states. Indeed, astrocyte-specific Cx43 conditional knockout mice exhibit abolished barrel-cortex LTP and deficits in whisker-guided sensorimotor learning, demonstrating that loss of inter- and intra-astrocytic coupling impairs memory-relevant circuit plasticity [183]. Genetic deletion of IP_3_R2, which abolishes global astrocytic Ca^2+^ elevations, leads to selective impairments in remote memory recall, without affecting initial learning or short-term recall, in tasks such as contextual fear conditioning and spatial navigation [184]. Thus, astrocytic Ca^2+^ signaling via IP_3_R2 is required for memory consolidation under specific behavioral conditions, especially for the formation and maintenance of long-term memories. Conversely, enhancing process-to-process coupling may improve memory persistence or improve recovery after neuronal damage [93]. Moreover, neuromodulatory signals such as norepinephrine [185] or acetylcholine [7,186] may act as bias terms, adjusting astrocytic calcium thresholds and thereby tuning memory encoding efficiency according to behavioral context. Together, emerging computational and biological evidence support a unifying view of astrocytes as hybrid analog–digital processors [93], analog in their graded Ca^2+^ signaling and digital in their discrete synaptic connectivity. Earlier models emphasized their role in homeostasis and modulation [78], while the NAAM model extends this framework to include hypothesized roles in computation and memory storage. Testing its predictions by mapping calcium attractor dynamics or manipulating interprocess coupling will be crucial to understanding how neuron–astrocyte cooperation underlies the brain’s extraordinary capacity, efficiency, and resilience.

## 5. Future Directions

Recent advances have greatly expanded our understanding of astrocytes, revealing their roles in microdomain-level signaling, multisynaptic integration, cell-wide calcium dynamics, and network-scale ensemble activity. Despite this conceptual progress, fundamental questions remain unresolved. First, the precise coding logic of astrocytic calcium signaling is still poorly understood. How do distinct spatial and temporal calcium patterns correspond to specific forms of plasticity, synaptic modulation, or behavioral output? Second, the stability and persistence of astrocytic ensembles over time are unknown. Do learning-associated astrocytes form stable long-term networks, or are they dynamically reconfigured during memory updating and forgetting? Third, the molecular determinants of information storage remain to be defined, including calcium signaling mechanisms that trigger transcriptional or epigenetic changes to stabilize astrocytic states.

Technological limitations currently constrain progress. Super-resolution modalities such as STED microscopy offer nanoscale reconstructions of astrocytic leaflets but face severe photobleaching, phototoxicity, and limited axial resolution, restricting their applicability to live imaging and volumetric dynamics [187]. Calcium imaging similarly shows large discrepancies between in vitro and in vivo measurements: organic dyes such as Fluo-4AM underestimate microdomain activity by up to 85–90% due to poor labeling of fine perisynaptic processes, while widefield and confocal imaging obscure subcellular compartmentalization by integrating signals across optical planes. Emerging optical solutions, particularly two-photon microscopy, address many of these limitations by reducing photodamage, improving depth penetration, and enabling reliable detection of Ca^2+^ signals across somata, branches, and microdomains in intact circuits. Two-photon light-sheet approaches reduce photobleaching further while supporting rapid volumetric imaging. Coupling these methods with membrane-targeted GECIs, process-specific reporters, voltage or pH indicators, spatial transcriptomics, and proteomic tools will be essential for mapping astrocytic computation across microdomains, leaflets, and ensembles during behavior.

Future progress will require coordinated integration of multiple approaches. At the molecular level, multi-scale analyses such as advanced imaging, ribosome profiling, proximity labeling, and epigenomic mapping will be needed to achieve a mechanistic understanding of how intracellular calcium regulates transcriptional and translational programs. A single method is insufficient to delineate how calcium dynamics propagate to gene expression, metabolic coupling, and gliotransmission to convert transient activity into durable memory. We therefore integrate proteomics with transcriptomics in a spatiotemporally aligned manner to capture expression programs together with the concurrent rewiring of protein networks [157,188,189]. At the cellular and circuit levels, optogenetic and chemogenetic manipulation, including ensemble-tagging strategies, will allow selective interrogation of astrocyte subpopulations during behavior. Computational modeling remains crucial for linking subcellular processes to systems-level outcomes, enabling quantitative predictions about learning capacity, information flow, and metabolic efficiency. Finally, incorporating astrocytic dynamics into artificial neural-network architectures may inspire new paradigms in adaptive and energy-efficient computing.

## 6. Conclusions

Astrocytes have moved from the periphery of neuroscience to a central position in discussions of cognition, plasticity, and memory. Across nanoscale microdomains, multi-synaptic leaflets, single-cell territories, and network-level ensembles, they provide computational capabilities that complement and extend fast neuronal signaling. Their slow calcium dynamics, adhesion-based synaptic organization, local translation, and metabolic flexibility together offer a multi-scale mechanism that may stabilize, contextualize, and integrate information across time.

Key messages emerging from this review are as follows:(1)**Astrocytes participate in memory formation through diverse mechanisms**, including parallel processing and integration of inputs from multiple synapses, the structural and functional regulation of synapses, circuit-level neuronal regulation and the encoding of memory within astrocyte ensembles. These processes operate hierarchically across microdomains, multi-synaptic leaflets, single-cell territories, and network-level ensembles, enabling astrocytes to couple local plasticity with large-scale circuit adaptation.(2)**Explaining the brain’s remarkable computational and mnemonic capabilities may require moving beyond neuron- and synapse-centric views to include astrocytic dynamics**, as increasingly supported by emerging experimental and computational frameworks. These models propose that slow, integrative calcium states within astrocytes could complement fast neuronal signaling by providing temporally extended, activity-silent forms of information storage. Incorporating astrocytes into memory theory may therefore help reconcile how the brain achieves high capacity, stability, and robustness with limited metabolic cost. The stability and robustness potentially conferred by such astrocytic memory codes along with possible metabolic efficiency gains remain theoretically compelling but require further experimental validation.

Clinically, disruptions in these astrocytic processes ranging from impaired Ca^2+^ signaling and altered gliotransmission to defective local translation, metabolic dysfunction, and abnormal synaptic adhesion are increasingly implicated in disorders such as Alzheimer’s disease, Parkinson’s disease, autism spectrum disorder, depression, epilepsy, and traumatic brain injury. Understanding how astrocytes shape plasticity across scales may therefore reveal new therapeutic targets, including modulators of astrocytic calcium dynamics, metabolic pathways, adhesion molecules, and local translational control.

Future progress will require integrating high-resolution imaging, cell-type-specific manipulation, computational modeling, and behavioral ensemble tagging to critically test whether and how astrocytes encode, store, and retrieve information. By uncovering the multicellular logic of neuron–astrocyte cooperation, we may not only reshape fundamental theories of memory and learning but also accelerate the development of treatments for cognitive disorders and inspire new paradigms in adaptive, energy-efficient computing.

## Figures and Tables

**Figure 1 cells-14-01936-f001:**
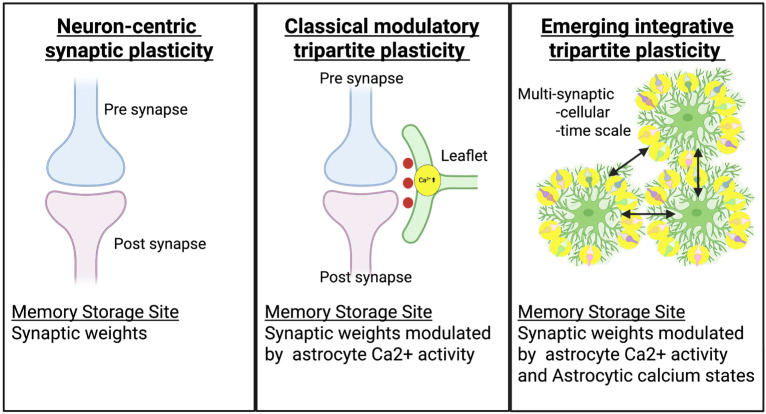
Comparison of the key components and memory storage sites across three conceptual models of synaptic plasticity.

**Figure 2 cells-14-01936-f002:**
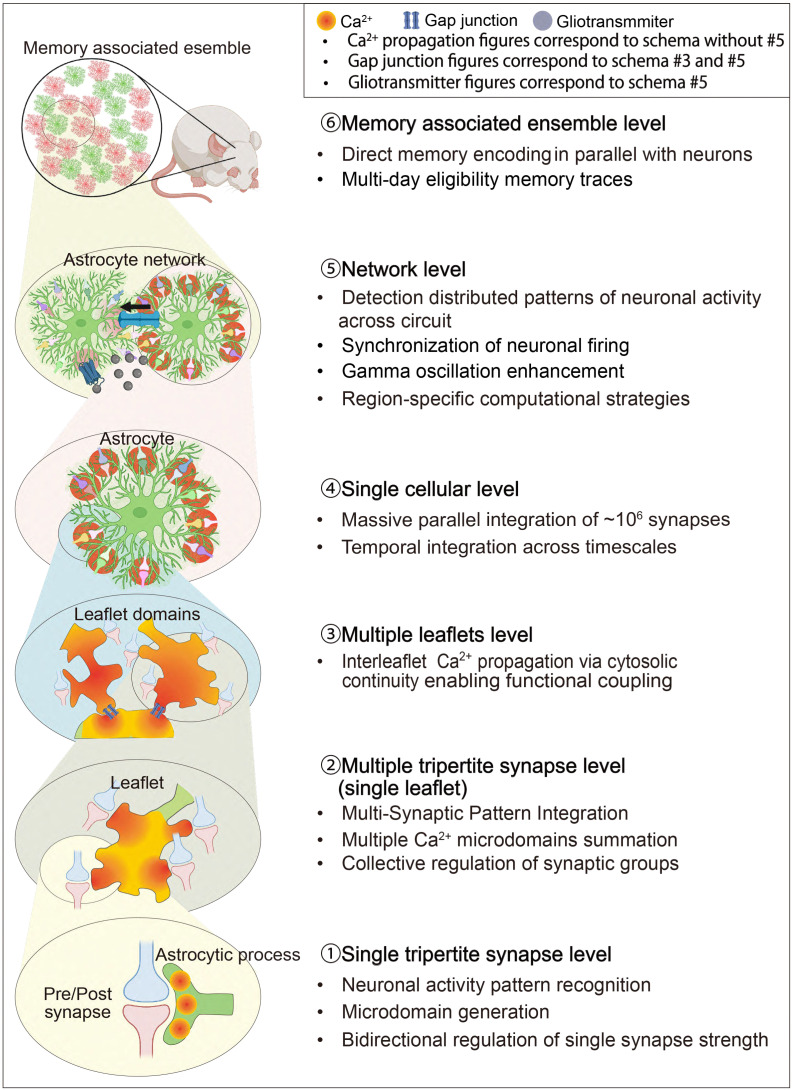
Hierarchical Structure of the Astrocyte-Integrated Framework.

**Table 1 cells-14-01936-t001:** Comparison of three frameworks.

Dimension	Neuron-Centric Synapse Plasticity	Classical Tripartite Synapse Plasticity	Emerging Integrative Tripartite Synapse Plasticity
Core Concept	Memory resides exclusively in synaptic weights between neurons; brain computation is purely neuronal	Astrocytes bidirectionally communicate with synapses; modulate synaptic transmission but do not store information	Astrocytes are computational units storing memories; neuron-astrocyte computational partnership
Memory Storage Site	Synaptic weights (connection strengths between neurons)	Synaptic weights dynamically modulated by astrocyte Ca^2+^ activity	Dual storage: synaptic weights + astrocytic Ca^2+^ state
Key Components	Presynaptic neuron + postsynaptic neuron	Presynaptic neuron + postsynaptic neuron + astrocytic leaflet (perisynaptic astrocyte process) single tripartite synapse	Hierarchical multi-scale tripartite coordination: from individual synapses to astrocyte leaflet domain networks implementing higher-order metaplasticity and circuit stabilization
Information Processing	Feed-forward and recurrent neural connections; pattern completion via attractor dynamics	Neurons activate astrocytes via neurotransmitters; astrocytes modulate synapses via gliotransmitter release	Parallel processing in neuron and astrocyte networks; astrocytes integrate information across 10^6^ synapses
Temporal Dynamics	Milliseconds (action potentials) to seconds (synaptic plasticity)	Milliseconds (neurons) to seconds/minutes (astrocyte Ca^2+^ waves)	Milliseconds to days: fast neuronal spikes/slow astrocyte Ca^2+^/multi-day molecular memory traces
Spatial Scale	Local synapses and neural circuit ensembles	Local tripartite synapse with limited spatial coordination between synapses	Multi-scale: microdomains (processes), meso-scale (single astrocyte tiles ~10^6^ synapses), macro-scale (astrocyte networks)
Astrocyte Role	Not included in computational models	Modulatory support role: sense neurotransmitters (glutamate, GABA), release gliotransmitters (ATP, D-serine)	Active computational partner: memory storage units, pattern recognition, attention-like gating, multi-day trace formation
Computational Capacity	Limited to synaptic weight matrix capacity	Slightly enhanced by astrocyte modulation of synaptic weight dynamics	Massively expanded: supralinear memory scaling enables exponential capacity growth with network size
Memory Scaling Law	Constant: M/N = constant (memories per neuron remains fixed as network grows)	Similar to neuron-centric: M/N ≈ constant (modulation does not fundamentally change scaling)	Supralinear: M/N grows with N (e.g., M ∝ N^3^ when astrocytes couple process pairs)
Synaptic Weight Control	Static or slowly changing via Hebbian/STDP rules	Dynamically modulated by astrocyte Ca^2+^ levels and gliotransmitter release timing	Online adaptive control: astrocytes continuously adjust effective synaptic weights based on network state
Network Architecture	Hopfield networks, attractor networks, recurrent neural networks	Extended Hopfield networks with astrocyte-mediated gliotransmission feedback loops	Dense Associative Memory, Modern Hopfield Networks; intermediate between DAMs and Transformers
Major Biological Basis (Neurons)	Spike generation, LTP/LTD, action potential propagation, synaptic transmission	Synaptic transmission + neurotransmitter receptor activation on astrocytes + bidirectional signaling	Neural activation essential for memory (engram hypothesis); astrocytes collaborate with neuronal ensembles
Major Biological Basis (Astrocytes)	Not applicable (astrocytes completely ignored in framework)	Ca^2+^ waves via IP3R2, gliotransmitter release (glutamate, ATP, D-serine), GPCR activation	Ca^2+^ microdomains in processes, process-process Ca^2+^ transport, multi-day molecular traces (IGFBP2, ADRB1 upregulation), ensemble formation
Time Period of Dominance	1980s–2010 (dominant paradigm)	1999–2020 (peak influence 2005–2015)	2016–present (accelerating 2020–2025)
Primary Advantages	Mathematical elegance, well-understood convergence properties, strong AI/ML connections, computational simplicity	Incorporates astrocyte biology, bidirectional neuron-glia signaling, explains gliotransmitter modulation effects	Explains brain’s massive memory capacity, multi-day stabilization, biologically detailed, supralinear scaling
Major Limitations	Ignores astrocytes entirely, lacks temporal dynamics beyond plasticity, limited memory capacity, static weights	Slow Ca^2+^ waves vs. fast synaptic events, passive modulation role, limited memory capacity enhancement	Complex parameter space, requires experimental validation of process-stored memories, molecular mechanisms incomplete

**Table 2 cells-14-01936-t002:** (**A**). Hierarchical Structure of the Astrocyte-Integrated Framework (MICRO-, MESO-, MACRO-SCALE). (**B**). Hierarchical Structure of the Astrocyte-Integrated Framework (NETWORK-, ENSEMBLE-SCALE).

** (A) **				
** Hierarchical Level **	** Structural Unit **	** Key Functional Properties **	** Memory Mechanism **	** Key References **
MICRO-SCALE (nanometer to micrometer)	IP_3_R-enriched ER fragments and Metabotropic receptors	Ca^2+^ signals are analog and graded with threshold gating.	IP_3_R-Ca^2+^ enables synapse-specific memory encoding; signal substrate	[13,92,95,96,97]
	Spatially confined Ca^2+^ zones	Microdomain dynamics; independent per-synapse processing; STDP gating	Synaptic weight encoding via pattern-specific Ca^2+^ dynamics. Parallel processing of ~100,000 synapses.	[6,7,8,14,32,79,81,84]
	Individual astrocyte leaflet	Ca^2+^-dependent mRNA localization; produces memory-linked proteins.	Long-term memory consolidation via local astrocyte protein synthesis	[12,14,98,99,100,101]
MESO-SCALE (single to few synapses)	Ultra-thin lamellar extension (leaflet)	Rapid Ca^2+^ transients (80–140 nM); single-synapse detection	Single-synapse memory encoding via Ca^2+^-gliotransmitter coupling	[7,14,45,46,96,102]
	Connexin43 gap junctions and luster of interconnected leaflets	Gap junction coupling; cytosolic continuity. ~10 synapses per leaflet; co-activated → merged Ca^2+^ waves	Coordinated plasticity across synapse clusters. Cooperative processing; collective memory traces	[14,40,41,42,103,104,105,106]
MACRO-SCALE (single astrocyte)	Hierarchical branching system	Hierarchical routing; integrates leaflet signals → soma coordination	Hierarchical memory organization across domains	[40,86,88,92,107]
	Complete astrocyte territorial domain	10,000–100,000 synapses per astrocyte,	Single astrocyte memory capacity for ~100,000 synapses	[10,40,94,108,109,110]
** (B) **				
** Hierarchical Level **	** Structural Unit **	** Key Functional Properties **	** Memory Mechanism **	** Key References **
NETWORK-SCALE (multi-astrocyte)	Locally coordinated tripartite clusters	Tripartite synapse coordination; ensemble plasticity	Ensemble-level memory trace formation	[3,4,14,44]
	Multi-astrocyte network (circuit-level)	Distributed computation; circuit pattern recognition	Circuit-level memory storage; E-I balance maintenance	[10,11,25,52,57,103,111]
	Oscillatory control hub	Theta-gamma synchronization; oscillatory power modulation	Theta-gamma coupling for episodic memories; oscillatory encoding	[89,112,113]
	Temporal integrator	Seconds-to-minutes integration; metaplasticity gating	Metaplasticity gating; experience-dependent plasticity windows	[6,8,25,88,109,114]
ENSEMBLE-SCALE (learning-activated)	Spatially clustered c-Fos+ astrocytes	NFIA-regulated; c-Fos co-expression with neurons	Direct memory encoding alongside engram neurons	[17,44,115,116,117]
	LAAs positioned near engram neurons	Physical proximity to engram neurons; bidirectional signaling	Engram stabilization; coordination with neuronal ensembles	[17,91]
	Multi-day molecular signatures	Multi-day molecular marks; adrenergic receptor upregulation; IGFBP2 storage	Long-term eligibility traces; resistance to decay	[91]
	Distributed ensembles (HC → Amy → PFC)	Multi-region coordination; HC → Amy → PFC progression during consolidation	Consolidation: rapid HC encoding → Amy emotional tagging → PFC storage	[17,44,85,91]

**Table 3 cells-14-01936-t003:** Quantitative summary of astrocytic structural units and calcium dynamics.

Scale	Structural Dimension	Ca^2+^ Signal Time Course	Synaptic Coverage/Functional Role	Representative Sources
Microdomain (node–shaft)	200–400 nm nodes; 50–200 nm shafts	Rise: 20–200 ms; Duration: 0.2–1.5 s	Single-synapse read–write signaling	[13,15]
Leaflet domain	~2–5 µm territory	Ca^2+^ events 200–800 ms	5–20 synapses per leaflet	[131]
Multi-leaflet region	10–20 µm	Ca^2+^ clustering 0.5–2 s	Integrates multisynaptic inputs	[1]
Single astrocyte territory	30–80 µm radius	Ca^2+^ waves 1–10 s	10^5^–10^6^ synapses per cell	[108,132]
Network level	mm-scale	Slow Ca^2+^ waves: 5–20 µm/s	Coordinates ensemble-level states	[105]

## Data Availability

Not applicable.

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
