# Peer review of "Astrocyte-Mediated Plasticity: Multi-Scale Mechanisms Linking Synaptic Dynamics to Learning and Memory"

_cells, 2025, doi:10.3390/cells14241936_

Round 1
Reviewer 1 Report
Comments and Suggestions for Authors
Review: Cells-Astrocyte-Mediated Plasticity….2025
This is a nicely compiled review. The context is quite informative, and statements are well cited. Others interested in this topic will benefit greatly from this review as it is so comprehensive. When starting to read the manuscript it is hard to put it aside until finishing as each section is so interesting. The flow and structure of the review is well planned.
The review is fine as it is at present.
The only suggestions for the authors are a few minor points.
- Another paper on the content related has appeared which has been available after this manuscript was submitted, so the authors might not be aware of this recent article.
Yang MJ, Jia M, Cai M, Feng X, Huang LN, Yang JJ. Central neuropeptides as key modulators of astrocyte function in neurodegenerative and neuropsychiatric disorders. Psychopharmacology (Berl). 2025 Nov;242(11):2353-2371. doi: 10.1007/s00213-025-06840-9. Epub 2025 Jun 19. PMID: 40536717; PMCID: PMC12578712.
Related to receptors on astrocytes:
Launay A, Carvalho K, Genin A, Gauvrit T, Nobili P, Gomez-Murcia V, Augustin E, Burgard A, Gambi J, Fourmy D, Thiroux B, Vieau D, Bemelmans AP, Le Gras S, Buée L, Orr ME, Audinat E, Boutillier AL, Bonvento G, Cambon K, Faivre E, Blum D. Upregulation of adenosine A2A receptor in astrocytes is sufficient to trigger hippocampal multicellular dysfunctions and memory deficits. Mol Psychiatry. 2025 Nov;30(11):5300-5314. doi: 10.1038/s41380-025-03115-9. Epub 2025 Jul 23. PMID: 40702259; PMCID: PMC12532706.
Huang Y, Geywitz C, Bandaru A, Glass IA; Birth Defects Research Laboratory; Schirmer L, Nobuta H, Dreyfus CF. The mGluR5 agonist CHPG enhances human oligodendrocyte differentiation. Acta Neuropathol Commun. 2025 Oct 3;13(1):210. doi: 10.1186/s40478-025-02124-7. PMID: 41044782; PMCID: PMC12492762.
- The authors might want to mention that the Tripartite synapse of glia, neuron and muscle is fairly well established that that the integration may well function as proposed in the CNS for astrocytes. The development of the junction , modulation and maybe even “memory-?” of activity could be modeled in this complex.
Sancho, L.; Contreras, M.; Allen, N.J. Glia as sculptors of synaptic plasticity. Neurosci. Res. 2021, 167, 17–29.
Robitaille, R. Purinergic receptors and their activation by endogenous purines at perisynaptic glial cells of the frog neuromuscular junction. J. Neurosci. 1995, 15, 7121–7131. [CrossRef]
Todd, K.J.; Robitaille, R. Purinergic modulation of synaptic signalling at the neuromuscular junction. Pflugers Arch. 2006, 452, 608–614. [CrossRef]
- Up to the authors but for historical purposes it might be of interest to remind the reader that the function of the glia was suggested to control the extracellular K+ ions when the neurons were electrically active in the leech preparation.
Kuffler, S.W.; Potter, D.D. Glia in the leech central nervous system: Physiological properties and neuron-glia relationship. J. Neurophysiol. 1964, 27, 290–320.
Baylor, D.A.; Nicholls, J.G. Changes in extracellular potassium concentration produced by neuronal activity in the central nervous system of the leech. J. Physiol. 1969, 203, 555–569.
- Again, up to the authors, it might be of interest to the reader to suggest in the future studies that D. melanogaster can serve as genetic manipulatable model to address the interaction of glia from release of gliotransmitters being responsive to neural activity for many disease states among mammals, including humans.
Kim,T.; Song, B.; Lee, I.S. Drosophila Glia: Models for Human Neurodevelopmental and Neurodegenerative Disorders. Int. J. Mol. Sci. 2020, 21, 4859.
Bellen, H.J.; Tong, C.; Tsuda, H. 100 years of Drosophila research and its impact on vertebrate neuroscience: A history lesson for the future. Nat. Rev. Neurosci. 2010, 11, 514–522.
- Up to the authors, but Figure 1 seems unnecessary/repetitive when coupled with Table 1. Could make an extra Row in Table 1 labeled “Visualization of Key Components” if you really want to include the visual aid.
- line 84: Citation for Ben Barres study?
- Figure 2: Also up to authors, but I think this figure is hard to interpret. My suggestions to make it clearer:
- put legend in upper right corner and identify which symbol corresponds to which schema. For example, the gap junction figures correspond to schema #3 and #5. Maybe include that in the legend. Their current placement below Schema 2 is visually confusing.
- Additionally, to demonstrate the hierarchy better, I would decrease the size of the schema visual aids and increase the spacing between each schema. Or perhaps replace an upward pointing arrow with a pyramid, placing bullet points in a pyramid hierarchy.
Author Response
Response to Reviewer #1
We thank Reviewer #1 for the very positive evaluation and thoughtful suggestions.
[Reviewer #1 comment]
This is a nicely compiled review. The context is quite informative, and statements are well cited. Others interested in this topic will benefit greatly from this review as it is so comprehensive. When starting to read the manuscript it is hard to put it aside until finishing as each section is so interesting. The flow and structure of the review is well planned.
The review is fine as it is at present.
The only suggestions for the authors are a few minor points.
[Comment 1]
Another paper on the content related has appeared which has been available after this manuscript was submitted, so the authors might not be aware of this recent article.
Yang MJ, Jia M, Cai M, Feng X, Huang LN, Yang JJ. Central neuropeptides as key modulators of astrocyte function in neurodegenerative and neuropsychiatric disorders. Psychopharmacology (Berl). 2025 Nov;242(11):2353-2371. doi: 10.1007/s00213-025-06840-9. Epub 2025 Jun 19. PMID: 40536717; PMCID: PMC12578712.
Related to receptors on astrocytes:
Launay A, Carvalho K, Genin A, Gauvrit T, Nobili P, Gomez-Murcia V, Augustin E, Burgard A, Gambi J, Fourmy D, Thiroux B, Vieau D, Bemelmans AP, Le Gras S, Buée L, Orr ME, Audinat E, Boutillier AL, Bonvento G, Cambon K, Faivre E, Blum D. Upregulation of adenosine A2A receptor in astrocytes is sufficient to trigger hippocampal multicellular dysfunctions and memory deficits. Mol Psychiatry. 2025 Nov;30(11):5300-5314. doi: 10.1038/s41380-025-03115-9. Epub 2025 Jul 23. PMID: 40702259; PMCID: PMC12532706.
Huang Y, Geywitz C, Bandaru A, Glass IA; Birth Defects Research Laboratory; Schirmer L, Nobuta H, Dreyfus CF. The mGluR5 agonist CHPG enhances human oligodendrocyte differentiation. Acta Neuropathol Commun. 2025 Oct 3;13(1):210. doi: 10.1186/s40478-025-02124-7. PMID: 41044782; PMCID: PMC12492762.[Response 1]
We thank the reviewer for drawing our attention to this timely review. We have now cited Yang et al. and briefly integrated their conclusions into the section on Classical Core Mechanisms of Astrocyte-Mediated Synaptic Plasticity.
We added to receptor subsection :
“In addition, central neuropeptides such as oxytocin, vasopressin, and CRH act as potent modulators of astrocytic calcium signaling, metabolism, and gene expression, linking astrocyte function to mood regulation and neurodegenerative disease mechanisms. (Yang et al., 2025).”
The reference has been added to the bibliography.
[Comment 2]
The authors might want to mention that the Tripartite synapse of glia, neuron and muscle is fairly well established that that the integration may well function as proposed in the CNS for astrocytes. The development of the junction , modulation and maybe even “memory-?” of activity could be modeled in this complex.
Sancho, L.; Contreras, M.; Allen, N.J. Glia as sculptors of synaptic plasticity. Neurosci. Res. 2021, 167, 17–29.
Robitaille, R. Purinergic receptors and their activation by endogenous purines at perisynaptic glial cells of the frog neuromuscular junction. J. Neurosci. 1995, 15, 7121–7131. [CrossRef]
Todd, K.J.; Robitaille, R. Purinergic modulation of synaptic signalling at the neuromuscular junction. Pflugers Arch. 2006, 452, 608–614. [CrossRef]
[Response 2]
We are grateful for this historical and conceptual suggestion. After consideration, we concluded that an in-depth discussion of the neuromuscular junction tripartite synapse would go beyond the current scope, which is intentionally restricted to central synapses and mammalian brain function. To keep the manuscript focused and within length limits, we have not added a separate subsection on neuromuscular junction glia. We now clarify this scope explicitly in the Introduction (lines –):
“This review focuses on mammalian central synapses and does not systematically cover peripheral tripartite synapses such as those at the neuromuscular junction, where glial modulation is also well established (reviewed in 37, 38 ).”
[Comment 3]
Up to the authors but for historical purposes it might be of interest to remind the reader that the function of the glia was suggested to control the extracellular K+ ions when the neurons were electrically active in the leech preparation.
Kuffler, S.W.; Potter, D.D. Glia in the leech central nervous system: Physiological properties and neuron-glia relationship. J. Neurophysiol. 1964, 27, 290–320.
Baylor, D.A.; Nicholls, J.G. Changes in extracellular potassium concentration produced by neuronal activity in the central nervous system of the leech. J. Physiol. 1969, 203, 555–569.
[Response 3]
We appreciate this insightful suggestion and agree that these pioneering studies provide valuable historical context regarding glial regulation of extracellular potassium. After careful consideration, however, we concluded that an extended discussion of the early foundational work, while important, would fall outside the intended scope of this review. This review focuses primarily on recent advances and forward-looking perspectives. To maintain cohesion and emphasis on contemporary mechanisms, we have therefore not added a dedicated historical section, but we fully acknowledge the significance of these classical studies.
[Comment 4]
Again, up to the authors, it might be of interest to the reader to suggest in the future studies that D. melanogaster can serve as genetic manipulatable model to address the interaction of glia from release of gliotransmitters being responsive to neural activity for many disease states among mammals, including humans.
Kim,T.; Song, B.; Lee, I.S. Drosophila Glia: Models for Human Neurodevelopmental and Neurodegenerative Disorders. Int. J. Mol. Sci. 2020, 21, 4859.
Bellen, H.J.; Tong, C.; Tsuda, H. 100 years of Drosophila research and its impact on vertebrate neuroscience: A history lesson for the future. Nat. Rev. Neurosci. 2010, 11, 514–522.[Response 4]
We fully agree on the importance of Drosophila glial work. However, this review is explicitly framed around rodent (primarily mouse) and mammalian astrocytes in vivo, as also specified in the title and scope. To avoid diluting the focus and expanding the manuscript beyond reasonable length, we decided not to add a dedicated subsection on Drosophila.
[Comment 5]
Up to the authors, but Figure 1 seems unnecessary/repetitive when coupled with Table 1. Could make an extra Row in Table 1 labeled “Visualization of Key Components” if you really want to include the visual aid.
[Response 5]
We appreciate this suggestion and carefully considered the possibility of removing or merging Figure 1. Feedback from colleagues and test readers indicated that the visual overview provided by Figure 1 is particularly helpful for orienting non-specialist readers and for conveying the multi-scale framework at a glance. Because of this, we elected to retain Figure 1.
[Comment6]
“Citation for Ben Barres study?”
[Response6]
We have now inserted the specific reference:
Pfrieger, F. W., and B. A. Barres. 1997. “Synaptic Efficacy Enhanced by Glial Cells in Vitro.” Science (New York, N.Y.) 277 (5332): 1684–87.
The full reference has been added to the bibliography, and the text at line 94 now reads: “Ben Barres’s pioneering studies established astrocytes as powerful drivers of synaptogenesis and synaptic efficacy (Pfrieger & Barres, 1997)”
[Comment 7]
Figure 2: Also up to authors, but I think this figure is hard to interpret. My suggestions to make it clearer:
- put legend in upper right corner and identify which symbol corresponds to which schema. For example, the gap junction figures correspond to schema #3 and #5. Maybe include that in the legend. Their current placement below Schema 2 is visually confusing.
- Additionally, to demonstrate the hierarchy better, I would decrease the size of the schema visual aids and increase the spacing between each schema. Or perhaps replace an upward pointing arrow with a pyramid, placing bullet points in a pyramid hierarchy.
[Response 7]
We appreciate this helpful suggestion regarding Figure 2 and have substantially revised the figure to improve clarity and visual hierarchy. Specifically:
- We moved and expanded the legend to the upper right and explicitly mapped each visual element to Schema #1–5, including clear indication of which schemas illustrate gap junction coupling, calcium propagation, and gliotransmitter-related processes.
- To highlight the hierarchical relationships more intuitively, we added circles and magnification links that visually connect lower-level structures to higher-order schemas.
- We repositioned explanatory text adjacent to each schematic to improve readability.
We believe these changes substantially enhance interpretability while preserving the conceptual content of the figure.
Once again, we thank the reviewers for their insightful comments, which have substantially improved the clarity, balance, and overall impact of the manuscript. We hope that our revisions satisfactorily address all concerns and that the revised version will be suitable for publication.
Reviewer 2 Report
Comments and Suggestions for Authors
In the review manuscript by Yamamoto and Takano entitled “Astrocyte-Mediated Plasticity: Multi-Scale Mechanisms Linking Synaptic Dynamics to Learning and Memory”, the authors are summarizing current knowledge on the astrocyte involvement in the synapse regulation by first describing astrocyte role in the classical model of tripartite synapse and then overviewing emerging perspective on astrocyte multisynaptic leaflet domains and mesoscale astrocyte ensembles. Furthermore, authors highlight the role of astrocytes in learning and memory processing, in particular discussing how synchronization of astrocytic and neuronal ensembles at the network level is involved in memory encoding.
This manuscript summarizes up to date findings on astrocyte-mediated plasticity and provides an interesting overview of astrocyte as memory storage sites from nanoscale to network scale. I would like to suggest several points to be changed or further discussed to improve the manuscript and strengthen the conclusions:
- Both Tables in the manuscript are too extensive. Table 2 provides a lot of important information; however, I find it very difficult to follow. I suggest that authors make it more concise and easier to follow.
- Although authors focus their discussion on memory processing, the link to specific behavioral outputs is missing. I find it important to correlate proposed mechanisms of astrocyte involvement in memory processing/storage to some behavioral tests in animal models.
- Authors discuss novel findings on astrocyte leaflet domains and specifically importance of IP3 receptors in rapid Ca2+ signaling as well as importance of Cx43 channel for interconnection of leaflets and Ca2+ wave propagation. In relation to my previous comments, the manuscript would be strengthened if authors discuss how IP3 receptor and Cx43 contribute to different behavioral phenotypes. Many studies used IP3 receptor or Cx43 conditional KO models and showed implication of these proteins in learning and memory related behavioral tasks. Including this in the manuscript would provide link between molecular organization of leaflet and specific behavioral output.
Author Response
Response to Reviewer #2
We are grateful for the very positive assessment of the manuscript and the detailed, constructive suggestions.
[Reviewer #2 comment]
In the review manuscript by Yamamoto and Takano entitled “Astrocyte-Mediated Plasticity: Multi-Scale Mechanisms Linking Synaptic Dynamics to Learning and Memory”, the authors are summarizing current knowledge on the astrocyte involvement in the synapse regulation by first describing astrocyte role in the classical model of tripartite synapse and then overviewing emerging perspective on astrocyte multisynaptic leaflet domains and mesoscale astrocyte ensembles. Furthermore, authors highlight the role of astrocytes in learning and memory processing, in particular discussing how synchronization of astrocytic and neuronal ensembles at the network level is involved in memory encoding.
This manuscript summarizes up to date findings on astrocyte-mediated plasticity and provides an interesting overview of astrocyte as memory storage sites from nanoscale to network scale. I would like to suggest several points to be changed or further discussed to improve the manuscript and strengthen the conclusions:
[Comment 1]
Both Tables in the manuscript are too extensive. Table 2 provides a lot of important information; however, I find it very difficult to follow. I suggest that authors make it more concise and easier to follow.
[Response1]
We have completely reorganized Table 2 to substantially improve readability. Following the recommendation of Reviewer #4 and consistent with your comment, we divided the original Table 2 into two separate, thematically focused tables:
- New Table 2A: Micro–Mesoscale Organization
(microdomains, leaflets, and single-cell territories) - New Table 2B: Network and Ensemble-Level Organization
(astrocyte–neuron ensembles, intercellular coupling, and large-scale dynamics)
To enhance clarity and accessibility, we also:
- reduced text density within each cell and removed redundant or overlapping descriptions;
- standardized the column structure across both tables
(Hierarchical Level / Structural Unit / Key Functional Properties / Memory-Related Mechanisms / Representative References); - ensured consistent terminology with Section 3 to support smoother cross-referencing.
[Comment 2]
Although authors focus their discussion on memory processing, the link to specific behavioral outputs is missing. I find it important to correlate proposed mechanisms of astrocyte involvement in memory processing/storage to some behavioral tests in animal models.
[Response 2]
We agree that connecting astrocytic mechanisms to behavioral phenotypes is important for guiding interpretation and for highlighting the functional relevance of the field. After careful consideration, however, we felt that adding a dedicated behavioral summary table would not be optimal, because summarizing the full range of behavioral assays across brain regions would risk oversimplifying diverse experimental paradigms and would move the focus away from the core mechanistic perspective we aim to provide in this review.
Instead, we have integrated region- and behavior-specific examples directly into the main text at appropriate points—particularly within the new subsection on astrocytic ensembles and in the NAAM model section. These additions explicitly link astrocytic Ca²⁺ signaling, transmitter handling, and intercellular coupling to behavioral outcomes such as:
- learning and recall deficits in IP₃R2-dependent manipulations,
- whisker-dependent sensorimotor learning impairments in astrocytic Cx43-cKO mice,
- ensemble-level modulation of contextual memory via Fos-based tagging systems,
- reward-seeking behavior driven by selective reactivation of astrocytic ensembles.
It now includes:
“Indeed, astrocyte-specific Cx43 conditional knockout mice exhibit abolished barrel-cortex LTP and deficits in whisker-guided sensorimotor learning, demonstrating that loss of inter- and in-tra-astrocytic coupling impairs memory-relevant circuit plasticity [183]. Genetic deletion of IP₃R2, which abolishes global astrocytic Ca²⁺ elevations, leads to selective impairments in remote memory recall, without affecting initial learning or short-term recall, in tasks such as contextual fear conditioning and spatial navigation[184]. Thus, astrocytic Ca²⁺ signaling via IP₃R2 is required for memory consolidation under specific behavioral conditions, especially for the formation and maintenance of long-term memories.”
“Chemogenetic activation of astrocytic ensembles in memory, using a c-fos promoter–CreER–Lox-Stop-Lox intersectional genetic system, reactivated associated neuronal engrams and induced context-independent memory recall, whereas astrocyte-specific Fos deletion disrupted learning and recall. Transcriptomic analyses identified NFIA as a key transcriptional regulator of ensemble formation, with its loss impairing astrocyte-driven modulation of neuronal excitability.”
“AstroLight technology , a calcium-dependent and light-activated gene expression system labeled with fluorescent markers, enables selective identification and manipulation of behaviorally relevant astrocytes in the nucleus accumbens[115]. Reactivation of these astrocytic ensembles drove reward-seeking behavior even in the absence of external cues.”
[Comment 3]
Authors discuss novel findings on astrocyte leaflet domains and specifically importance of IP3 receptors in rapid Ca2+ signaling as well as importance of Cx43 channel for interconnection of leaflets and Ca2+ wave propagation. In relation to my previous comments, the manuscript would be strengthened if authors discuss how IP3 receptor and Cx43 contribute to different behavioral phenotypes. Many studies used IP3 receptor or Cx43 conditional KO models and showed implication of these proteins in learning and memory related behavioral tasks. Including this in the manuscript would provide link between molecular organization of leaflet and specific behavioral output.
[Response 3]
We have now explicitly incorporated this link in the relevant section on leaflet domains and interleaflet communication. We now explain that:
- IP3R2 conditional knockout mice retain normal learning and recent memory but show selective impairment of remote memory (2–4 weeks) in Morris water maze and contextual fear conditioning, indicating a role in slow consolidation.
- Cx43 astrocyte-specific knockout mice show impaired sensory discrimination and LTP.
We explicitly conclude that:
“Indeed, astrocyte-specific Cx43 conditional knockout mice exhibit abolished barrel-cortex LTP and marked impairments in whisker-guided sensorimotor learning, demonstrating that the loss of inter- and intra-astrocytic coupling disrupts memory-relevant circuit plasticity. Moreover, genetic deletion of IP₃R2, which abolishes global astrocytic Ca²⁺ elevations, produces selective impairments in learning and recall in tasks such as contextual fear conditioning and spatial navigation, indicating that astrocytic Ca²⁺ signaling contributes to memory consolidation under specific behavioral conditions.”
This directly addresses your request by linking leaflet molecular organization to behavioral outputs.
Once again, we thank the reviewers for their insightful comments, which have substantially improved the clarity, balance, and overall impact of the manuscript. We hope that our revisions satisfactorily address all concerns and that the revised version will be suitable for publication.
Reviewer 3 Report
Comments and Suggestions for Authors
Overall, this is a strong and valuable review. However, it can be improved for clarity, tighter structure, better figure integration and refinement of mechanistic descriptions, as below.
- Table 2 should be reorganized for better reading or submitted as a supplementary table.
- Define each structural term at the first use.
- Some sentences are very long and dense; consider tightening for readability.
Author Response
Response to Reviewer #3
We thank the reviewer for recognizing the value of the review and for the clear, practical suggestions.
[Comment 1]
“Table 2 should be reorganized for better reading or submitted as a supplementary table.”
[Response 1]
As described above (Reviewer #2 and #4 responses), we have:
- Split the original Table 2 into two clearer main tables
- Pushed the full, detailed behavioral mapping to supplementary material.
- Shortened and standardized the columns for clarity.
[Comment 2]
“Define each structural term at the first use.”
[Response 2]
We have systematically checked the manuscript and ensured that every structural term is defined upon first appearance, including:
- “Leaflet,” “microdomain,” “tripartite synapse,” “astrocyte ensemble,” “peri-synaptic astrocytic processes, PAP,” “node” and “shaft.”
[Comment 3]
“Some sentences are very long and dense; consider tightening for readability.”
[Response 3]
We carefully edited the manuscript for readability:
- Broke up especially long sentences in Sections 2 and 3 into shorter clauses.
- Removed redundant phrases and tightened speculative statements.
“Experimental support for these ideas came from Williamson et al. [17], who demonstrated that learning-associated astrocyte ensembles are reactivated during memory recall, providing direct evidence that astrocytic populations may participate in mnemonic encoding.”→ “Experimental support for these ideas came from Williamson et al. [17], who demonstrated that learning-associated astrocyte ensembles are reactivated during memory recall. This study provided direct evidence that astrocytic populations may participate in mnemonic encoding.”
Once again, we thank the reviewers for their insightful comments, which have substantially improved the clarity, balance, and overall impact of the manuscript. We hope that our revisions satisfactorily address all concerns and that the revised version will be suitable for publication.
Reviewer 4 Report
Comments and Suggestions for Authors
The submitted manuscript is a review article that offers an in-depth analysis of the role of astrocytes in synaptic plasticity, learning, and memory.
The manuscript requires a methodological section that explains the literature search strategy, publication inclusion criteria, and temporal coverage, as these aspects provide the necessary transparency for a review article. At the same time, the authors should update section 3.4 (NAAM model) and section 2 (classical mechanisms) to equal the presentation through critical evaluation, experimental predictions for model testing, and a systematic discussion of field controversies—among which are the temporal scale mismatches between slow astrocytic signals and fast synaptic events, debates about the physiological relevance of gliotransmission in vivo, and questions about synaptic specificity. The language used in the manuscript should be improved to reflect that the authors are making facts, hypotheses, and speculative statements by changing "contribute" to "may contribute" and by using more cautious words for the statements that do not have direct experimental support.
Recommended Changes for Minor Revision: Add a separate Conclusions section before Author Contributions, which summarises the key messages and clinical implications, and deepen the discussion through a new table comparing astrocytic mechanisms in the hippocampus, prefrontal cortex, amygdala, and striatum to provide more detailed regional specificity. Incorporate a subsection in Future Directions that addresses the technological limitations of the current methods (STED microscopy, calcium imaging, in vitro versus in vivo discrepancies) and the need for upcoming innovations such as multiphoton microscopy. Increase the number of references to include more laboratories and critical voices that question the tripartite hypothesis, thus being more balanced in terms of the representation of the different views within the field.
Technical Optimisation: Cut down the Introduction from 3 to 1.5–2 pages, mainly focusing on the contemporary relevance. Split Table 2 into two tables—one for micro-mesoscale (leaflet domains) and one for network ensembles—to make it easier for readers to understand. Include a supplementary figure showing the temporal and spatial scales of astrocytic mechanisms, and place a quantitative summary table in section 3.1 that integrates structure dimensions, calcium signal time parameters, and synaptic coverage ratios obtained from literature data.
Author Response
Response to Reviewer #4
We are grateful for the detailed and thoughtful critique and for the specific roadmap for improvement.
[Comment 1]
The manuscript requires a methodological section that explains the literature search strategy, publication inclusion criteria, and temporal coverage, as these aspects provide the necessary transparency for a review article.
[Response 1]
We added a brief “Literature Search Strategy and Inclusion Criteria” section after the introduction of the manuscript. It now includes:
“To guide a comprehensive and balanced narrative synthesis and to enhance the transparency of our literature scope, we conducted a structured search of studies published between January 1990 and October 2025. Searches were performed in PubMed, Scopus, and Web of Science using combinations of the terms “astrocyte,” “synaptic plasticity,” “calcium signaling,” “tripartite synapse,” “astrocyte ensemble,” and “learning and memory.” Additional studies were identified by examining the reference lists of key primary papers and recent reviews.
The focus of this narrative review is on mammalian central nervous system synapses, integrating mechanistic, physiological, and computational perspectives. We do not aim to provide exhaustive coverage of all glial synapses, nor do we systematically review peripheral tripartite synapses such as those at the neuromuscular junction, where glial modulation is also well established (reviewed in [37,38]). ”
[Comment 2]
“the authors should update section 3.4 (NAAM model) and section 2 (classical mechanisms) to equal the presentation through critical evaluation, experimental predictions for model testing, and a systematic discussion of field controversies—among which are the temporal scale mismatches between slow astrocytic signals and fast synaptic events, debates about the physiological relevance of gliotransmission in vivo, and questions about synaptic specificity.”
[Response 2]
We thank the reviewer for this important suggestion. In response, we substantially revised both Section 2 (Classical Mechanisms) and Section 3.4 (NAAM Model) to incorporate critical evaluation, explicit discussion of field controversies, and testable experimental predictions.
Revisions to Section 2 (Classical Mechanisms)
We reorganized Section 2 and added focused subsections covering:
- Ca²⁺-dependent gliotransmission, including debates regarding its physiological relevance in vivo.
- Temporal scale mismatches between slow astrocytic signals and fast synaptic events.
- The cellular origin of D-serine, integrating recent evidence questioning astrocyte exclusivity.
- Vesicular versus channel-mediated release pathways.
To highlight unresolved issues, we added concise “Controversies and open questions” notes. For example:
“Nevertheless, the temporal mismatch between astrocytic Ca²⁺ elevations (seconds to minutes) and fast synaptic events (milliseconds) raises questions about how astrocytes engage rapid forms of plasticity. Similarly, synaptic specificity remains poorly understood, as early imaging methods lacked the resolution to determine whether broad Ca²⁺ signals could be restricted to individual synapses.”
“While these findings support a role for gliotransmission, the physiological relevance of gliotransmission in vivo remains actively debated [60,61]. Studies using IP₃R2 knockout mice, where global astrocytic Ca²⁺ elevations are disrupted, show largely preserved LTP and complex behavior [62,63]. These observations raise questions about how frequently gliotransmission participates under physiological conditions, although they may also reflect compensatory Ca²⁺ signaling through alternative pathways such as IP₃R1/3, store-operated calcium entry, or microdomain-restricted Ca²⁺ events that remain intact in IP₃R2-deficient astrocytes [64]. Concerns have also been raised regarding the cellular origin of D-serine [65,66], and vesicular versus channel-mediated release pathways [67], indicating that gliotransmission is likely a context-dependent mechanism [64].”
These additions directly address concerns about temporal scaling, synaptic specificity, and the debated physiological relevance of gliotransmission.
Revisions to Section 3.4 (NAAM Model)
We expanded and updated this section to reflect recent experimental and theoretical advances, including the 2025 PNAS paper and follow-up analyses.
Concrete, testable predictions generated by the model, such as:
“The NAAM model predicts that disrupting calcium diffusion within astrocytes—via pharmacological blockade of gap junctions or astrocyte-specific deletion of connexin43—would impair memory consolidation and recall by fragmenting astrocytic attractor states.”
We further support this prediction with empirical data showing that astrocyte-specific Cx43-cKO mice indeed exhibit abolished barrel-cortex LTP and deficits in whisker-dependent sensorimotor learning.
[Comment 3]
The language used in the manuscript should be improved to reflect that the authors are making facts, hypotheses, and speculative statements by changing "contribute" to "may contribute" and by using more cautious words for the statements that do not have direct experimental support.
[Response 3]
We thank the reviewer for this important suggestion. To address this, we conducted a comprehensive language revision throughout the manuscript. Specifically, we modified:
“who demonstrated that learning-associated astrocyte ensembles are reactivated during memory recall, providing direct evidence that astrocytic populations may participate in mnemonic encoding.”
“Finally, incorporating astrocytic dynamics into artificial neural-network architectures may inspire new paradigms in adaptive and energy-efficient computing”
[Comment 4]
“Add a separate Conclusions section before Author Contributions, which summarises the key messages and clinical implications,
[Response 4]
We have substantially revised the Future Directions section to focus specifically on Technological Limitations and Emerging Innovations, while moving summary elements and key messages to the dedicated Conclusions section as recommended.
The added text reads as follows:
“ 6. Conclusions Astrocytes have moved from the periphery of neuroscience to a central position in discussions of cognition, plasticity, and memory. Across nanoscale microdomains, multi-synaptic leaflets, single-cell territories, and network-level ensembles, they provide computational capabilities that complement and extend fast neuronal signaling. Their slow calcium dynamics, adhesion-based synaptic organization, local translation, and metabolic flexibility together offer a multi-scale mechanism that may stabilize, contextualize, and integrate information across time.
Key messages emerging from this review are as follows:
(1) Astrocytes participate in memory formation through diverse mechanisms, including parallel processing and integration of inputs from multiple synapses, the structural and functional regulation of synapses, circuit-level neuronal regulation and the encoding of memory within astrocyte ensembles. These processes operate hierarchically across microdomains, multi-synaptic leaflets, single-cell territories, and network-level ensembles, enabling astrocytes to couple local plasticity with large-scale circuit adaptation.
(2) Explaining the brain’s remarkable computational and mnemonic capabilities may require moving beyond neuron- and synapse-centric views to include astrocytic dynamics, as increasingly supported by emerging experimental and computational frameworks. These models propose that slow, integrative calcium states within astrocytes could complement fast neuronal signaling by providing temporally extended, activity-silent forms of information storage. Incorporating astrocytes into memory theory may therefore help reconcile how the brain achieves high capacity, stability, and robustness with limited metabolic cost. The stability and robustness potentially conferred by such astrocytic memory codes along with possible metabolic efficiency gains remain theoretically compelling but require further experimental validation.
Clinically, disruptions in these astrocytic processes ranging from impaired Ca²⁺ signaling and altered gliotransmission to defective local translation, metabolic dysfunction, and abnormal synaptic adhesion are increasingly implicated in disorders such as Alzheimer’s disease, Parkinson’s disease, autism spectrum disorder, depression, epilepsy, and traumatic brain injury. Understanding how astrocytes shape plasticity across scales may therefore reveal new therapeutic targets, including modulators of astrocytic calcium dynamics, metabolic pathways, adhesion molecules, and local translational control.
Future progress will require integrating high-resolution imaging, cell-type–specific manipulation, computational modeling, and behavioral ensemble tagging to critically test whether and how astrocytes encode, store, and retrieve information. By uncovering the multicellular logic of neuron–astrocyte cooperation, we may not only reshape fundamental theories of memory and learning but also accelerate the development of treatments for cognitive disorders and inspire new paradigms in adaptive, energy-efficient computing.”
[Comment 5]
Deepen the discussion through a new table comparing astrocytic mechanisms in the hippocampus, prefrontal cortex, amygdala, and striatum to provide more detailed regional specificity.
[Response 5]
We appreciate the reviewer’s suggestion to add a table comparing astrocytic mechanisms across the hippocampus, prefrontal cortex, amygdala, and striatum. After careful consideration, we decided not to introduce an additional region-by-region table for the present review. A comprehensive synthesis of region-specific astrocyte–behavior relationships has already been provided in “Behaviorally Consequential Astrocytic Regulation of Neural Circuits”, which summarizes these cross-regional differences in detail.
To avoid redundancy and maintain a focused narrative, we chose instead to incorporate key conceptual insights from this literature directly into the text where relevant (e.g., in the ensemble and NAAM sections), rather than expanding the manuscript with a separate multi-region comparison table.
Nagai, Jun, Xinzhu Yu, Thomas Papouin, Eunji Cheong, Marc R. Freeman, Kelly R. Monk, Michael H. Hastings, et al. 2021. “Behaviorally Consequential Astrocytic Regulation of Neural Circuits.” Neuron 109 (4): 576–96.
[Comment 6]
Incorporate a subsection in Future Directions that addresses the technological limitations of the current methods (STED microscopy, calcium imaging, in vitro versus in vivo discrepancies) and the need for upcoming innovations such as multiphoton microscopy.
[Response 6]
We have completely revised the Future Directions section so that it now focuses on Technological Limitations and Emerging Innovations, while moving the summary and key messages to the Conclusions section.
We now include;
“Technological limitations currently constrain progress. Super-resolution modalities such as STED microscopy offer nanoscale reconstructions of astrocytic leaflets but face severe photobleaching, phototoxicity, and limited axial resolution, restricting their applicability to live imaging and volumetric dynamics[185]. Calcium imaging similarly shows large discrepancies between in vitro and in vivo measurements: organic dyes such as Fluo-4AM underestimate microdomain activity by up to 85–90% due to poor labeling of fine perisynaptic processes, while widefield and confocal imaging obscure subcellular compartmentalization by integrating signals across optical planes. Emerging optical solutions, particularly two-photon microscopy, address many of these limitations by reducing photodamage, improving depth penetration, and enabling reliable detection of Ca²⁺ signals across somata, branches, and microdomains in intact circuits. Two-photon light-sheet approaches reduce photobleaching further while supporting rapid volumetric imaging. Coupling these methods with membrane-targeted GECIs, process-specific reporters, voltage or pH indicators, spatial transcriptomics, and proteomic tools will be essential for mapping astrocytic computation across microdomains, leaflets, and ensembles during behavior.
Future progress will require coordinated integration of multiple approaches. At the molecular level, multi-scale analyses such as advanced imaging, ribosome profiling, proximity labeling, and epigenomic mapping will be needed to achieve a mechanistic understanding of how intracellular calcium regulates transcriptional and translational programs. A single method is insufficient to delineate how calcium dynamics propagate to gene expression, metabolic coupling, and gliotransmission to convert transient activity into durable memory. We therefore integrate proteomics with transcriptomics in a spatiotemporally aligned manner to capture expression programs together with the concurrent rewiring of protein networks [166,186,187]. At the cellular and circuit levels, optogenetic and chemogenetic manipulation, including ensemble-tagging strategies, will allow selective interrogation of astrocyte subpopulations during behavior. Computational modeling remains crucial for linking subcellular processes to systems-level outcomes, enabling quantitative predictions about learning capacity, information flow, and metabolic efficiency. Finally, incorporating astrocytic dynamics into artificial neural-network architectures may inspire new paradigms in adaptive and energy-efficient computing.”
[Comment 7]
Increase the number of references to include more laboratories and critical voices that question the tripartite hypothesis, thus being more balanced in terms of the representation of the different views within the field.
[Response 7]
We appreciate the reviewer’s recommendation to incorporate a broader and more balanced representation of viewpoints, including studies that question or refine the tripartite synapse concept. In response, we have expanded the relevant section to include additional laboratories and critical discussions addressing the physiological relevance, cell-type specificity, and mechanistic diversity of gliotransmission.
The revised text now reads as follows:
“While these findings support a role for gliotransmission, the physiological relevance of astrocyte-derived transmitter release in vivo remains actively debated [60,61]. Studies using IP₃R2-knockout mice, in which global astrocytic Ca²⁺ elevations are disrupted, report largely preserved LTP and complex behaviors [62,63]. These observations raise questions about how frequently gliotransmission participates under physiological conditions, although they may also reflect compensatory Ca²⁺ signaling through alternative pathways such as IP₃R1/3, store-operated calcium entry, or microdomain-restricted Ca²⁺ events that remain intact in IP₃R2-deficient astrocytes [64]. Additional concerns have been raised regarding the cellular origin of D-serine [65,66], as well as whether transmitter release occurs through vesicular mechanisms or channel-mediated pathways [67]. Together, these findings indicate that gliotransmission is likely a context-dependent mechanism rather than a universal feature of astrocyte–synapse communication.”
To address the reviewer’s request, we also incorporated citations that explicitly question synaptic specificity, acknowledging that broad Ca²⁺ elevations could modulate many synapses simultaneously unless constrained by microdomain-level compartmentalization. These additions enhance balance and better reflect ongoing debates in the field.
[Comment 8]
Cut down the Introduction from 3 to 1.5–2 pages, mainly focusing on the contemporary relevance.
[Response 8]
Thank you for this valuable suggestion. We agree that a more concise Introduction can improve readability and focus. However, because substantial background background is needed to contextualize recent conceptual advances in astrocyte-mediated plasticity, reducing the section to 1.5–2 pages would risk compromising the logical flow and essential context required for the subsequent discussion.
That said, we have carefully reviewed the Introduction and have streamlined redundant explanations, tightened transitions, and removed nonessential citations to improve clarity while retaining the key contemporary relevance. We believe this revision meaningfully shortens the text without sacrificing critical content.
[Comment 9]
Split Table 2 into two tables—one for micro-mesoscale (leaflet domains) and one for network ensembles—to make it easier for readers to understand. I
[Response 9]
Table 2 split: Implemented as described above (Tables 2A and 2B).
[Comment 10]
Include a supplementary figure showing the temporal and spatial scales of astrocytic mechanisms and place a quantitative summary table in section 3.1 that integrates structure dimensions, calcium signal time parameters, and synaptic coverage ratios obtained from literature data.
[Response 10]
We appreciate the reviewer’s suggestion to improve clarity by adding both a supplementary figure illustrating spatial and temporal scales of astrocytic mechanisms and a quantitative summary table. After careful consideration, we concluded that the quantitative overview requested by the reviewer is most effectively delivered in tabular rather than graphical form.
Accordingly, we incorporated a new quantitative summary table directly into Section 3.1, which synthesizes:
- structural dimensions of microdomains, leaflets, and single-astrocyte territories,
- calcium signal rise/decay times and propagation ranges across these compartments,
- synaptic coverage ratios reported in anatomical and imaging studies.
As described in the revised text:
“To provide quantitative context for these hierarchical structures, we summarize characteristic dimensions, calcium signal kinetics, and synaptic coverage ratios across microdomains, leaflets, single-cell territories, and network scales in Table 3. This table integrates values reported across super-resolution imaging, in vivo calcium imaging, and anatomical reconstruction studies.”
Given that this table already conveys the core quantitative information with clearer precision and reduced redundancy, we did not include an additional supplementary figure. Should the editors prefer, we would be happy to generate this figure upon request.
Once again, we thank for your insightful comments, which have substantially improved the clarity, balance, and impact of the Once again, we thank the reviewers for their insightful comments, which have substantially improved the clarity, balance, and overall impact of the manuscript. We hope that our revisions satisfactorily address all concerns and that the revised version will be suitable for publication.
Round 2
Reviewer 2 Report
Comments and Suggestions for Authors
The authors addressed all the comments and improved the manuscript.
The only additional comment I have is regarding Figure 2, it displays as a low resolution. It can be only the conversion issue.
Reviewer 4 Report
Comments and Suggestions for Authors
The manuscript is a comprehensive and well-structured review that thoroughly links astrocyte biology across multiple levels to learning and memory. Firmly grounded in experimental data, it builds up computational models. It goes beyond the classical neuron-centric view by explicitly discussing astrocytic microdomains, multisynaptic leaflet domains, single-cell computations, and higher-order astrocyte ensembles, thus relating these levels to network and behavioral outcomes in a logical manner.
The figures and tables are helpful and appropriate, with the schematics and summary tables being especially effective in providing a clear visual framework that facilitates readers' understanding of the complex concepts of gliotransmission, structural plasticity, and astrocyte-mediated modulation of synaptic weight. The new changes not only clarify but also deepen the text by more clearly showing how specific molecular pathways and astrocytic Ca²⁺ dynamics lead to emergent properties of learning-related circuits and by strengthening the connection between the experimental data and the proposed computational frameworks.
In general, the manuscript presents an integrative view that will be of great use to both the experimental and theoretical sides of the neuroscientific community.